# Endoplasmic Reticulum-Mitochondria Crosstalk in Fuchs Endothelial Corneal Dystrophy: Current Status and Future Prospects

**DOI:** 10.3390/ijms26030894

**Published:** 2025-01-22

**Authors:** Anisha Kasi, William Steidl, Varun Kumar

**Affiliations:** 1Eye and Vision Research Institute, Department of Ophthalmology, Icahn School of Medicine at Mount Sinai, 1 Gustave L. Levy Pl, New York, NY 10029, USA; anisha.kasi@icahn.mssm.edu (A.K.); william.steidl@icahn.mssm.edu (W.S.); 2Department of Pharmacological Sciences, Icahn School of Medicine at Mount Sinai, New York, NY 10029, USA

**Keywords:** Fuchs, ER stress, mitochondrial stress, crosstalk, MAMs

## Abstract

Fuchs endothelial corneal dystrophy (FECD) is a progressive and debilitating disorder of the corneal endothelium (CE) that affects approximately 4% of individuals over the age of 40. Despite the burden of the disease, the pathogenesis of FECD remains poorly understood, and treatment options are limited, highlighting the need for deeper investigation into its underlying molecular mechanisms. Over the past decade, studies have indicated independent contributions of endoplasmic reticulum (ER) and mitochondrial stress to the pathogenesis of FECD. However, there are limited studies suggesting ER-mitochondria crosstalk in FECD. Recently, our lab established the role of chronic ER stress in inducing mitochondrial dysfunction for corneal endothelial cells (CEnCs), indicating the existence of ER-mitochondria crosstalk in FECD. This paper aims to provide a comprehensive overview of the current understanding of how ER and mitochondrial stress contribute to FECD pathogenesis. The paper also reviews the literature on the mechanisms of ER-mitochondria crosstalk in other diseases relevant to FECD.

## 1. Fuchs Burden

Fuchs endothelial corneal dystrophy (FECD) is a bilateral, genetically heterogeneous [1], age-related degenerative [2] and female-prevalent disease [3,4] of the corneal endothelial cells (CEnCs), affecting ~1–4% of the US population above age 40 [5]. FECD poses a significant annual burden, both in human suffering and in medical costs. For instance, the 2019 prevalence of FECD in the United States Medicare population was about 285,000 patients (1.12%), with the national count estimated at 591,000 patients [5]. The economic burden of FECD in the Medicare age group was found to be about USD 291.6 million in 2019, up from USD 244 million in 2013 [5]. Currently, the mechanisms underlying FECD pathogenesis are not well understood, and corneal transplantation is the only treatment for this disorder [6]. While transplantation can restore vision, endothelial keratoplasty for the treatment of FECD carries the risk of developing complications like glaucoma [7] or steroid dependence [8], indicating a need for non-invasive medical therapies [9]. Endothelial keratoplasty also poses a significant economic burden on FECD patients, with post-surgical patients spending about USD 23,000 on eye-related medical costs versus about USD 1400 by non-surgical patients in the 12 months after enrollment, as shown in a 2019 study [9]. Given the pitfalls of surgical treatments, there is a need to develop new pharmacological interventions, for which molecular mechanisms underlying FECD pathogenesis must be understood in detail.

## 2. Fuchs Pathogenesis

In FECD, CEnCs display various chronic intracellular stresses, including oxidative, mitochondrial, and ER stress [10]. Of these, chronic ER and mitochondrial stress are considered the major contributors to CEnC apoptosis in FECD. However, the underlying molecular mechanisms of the specific ER and mitochondrial stress pathways that contribute to CEnC apoptosis remain elusive. ER and mitochondria interact primarily at mitochondrial-associated ER membranes (MAMs), which form contact sites and function as signaling hubs (calcium, lipid regulation) [11] that regulate many cellular functions [12]. Despite this, there have been no studies on MAMs in FECD. The following unknown questions are related to ER, mitochondrial stress, and MAMs in FECD.

(a)It remains unknown whether MAM disruptions are primary or secondary factors contributing to Fuchs’s pathophysiology.(b)How disrupted MAMs impair MAM proteins and their functions is also unknown.(c)Do MAM proteins mediate differential effects on ER or mitochondrial stress pathways that lead to CEnC apoptosis in FECD?

Moreover, there are also no pharmacological treatments for FECD that target the ER, mitochondria, or MAMs. Thus, there is an unmet need to understand the mechanisms of CEnC degeneration/apoptosis with respect to ER, mitochondria, and MAMs so that pharmacological therapies for FECD can be developed.

## 3. ER Stress and Corneal Endothelial Cells

In our lab’s Qureshi et al. 2023 paper, we demonstrated that tunicamycin-induced chronic ER stress led to the activation of PERK-ATF4-CHOP in an immortalized human corneal endothelial cell line (HCEnC-21T) [13]. Supporting these findings, Okumura et al. showed that Thapsigargin-induced ER stress led to increased activation of the PERK-ATF4-CHOP pathway in the human normal control cell line [14]. Together, these findings support the role of chronic UPR/ER stress, specifically the PERK-ATF4-CHOP pathway, in CEnCs. Furthermore, Okumura et al. showed that knockdown of CHOP led to decreased expression of cleaved caspase 9 under Thapsigargin-induced chronic ER stress in immortalized normal human corneal endothelial cell lines [14]. Recently, we showed that knockdown of ATF4 led to improved CEnC survival and decreased activation of caspases (cleaved caspase 3 and 9) under chronic ER stress [15]. These studies support the direct role of the PERK-ATF4-CHOP pathway in the CEnC apoptosis. The role of other UPR/ER stress pathways, i.e., IRE and ATF6, in CEnC apoptosis is still unclear. We reported increased expression of XBP1 upon tunicamycin-induced chronic ER stress in a normal 21T cell line [13], consistent with reports of IRE-XBP1 activation after Thapsigargin-induced ER stress [14]. These studies show the involvement of the IRE-XBP1 pathway in CEnC apoptosis. However, its direct effect on CEnC apoptosis remains elusive. Similar to the IRE-XBP1 pathway, we reported increased activation of cleaved ATF6 in the 21T cell line upon tunicamycin-induced chronic ER stress [13], with similar findings reported by Okumura’s group [14]. However, the direct role of the ATF6 pathway in CEnC apoptosis is still unclear.

## 4. ER Stress and FECD

ER stress engages the unfolded protein response (UPR), an adaptive reaction that restores protein-folding homeostasis and maintains cell viability under various pathological conditions. However, unresolved ER stress impairs protein homeostasis, triggering sustained activation of the UPR, which leads to apoptosis. There are three molecular pathways of the UPR [16], each of which is initiated by activation of one of three molecular receptors in the ER membrane: Protein kinase RNA-like ER kinase (PERK), Inositol-requiring enzyme 1 (IRE1), and Activating transcription factor (ATF6). One of the important, well-studied UPR pathways in FECD is the PERK-eIF2-ATF4-CHOP pathway. Briefly, PERK phosphorylates eukaryotic initiation factor 2 (eIF2), which results in inhibition of global protein synthesis and sustained upregulation of activating transcription factor (ATF4) and CCAAT-enhancer-binding protein homologous protein (CHOP) under chronic ER stress. Engler et al. using immunohistochemistry demonstrated that there was an increase in the expression of eIF2α and CHOP proteins in human FECD tissues compared to non-diseased corneal tissues, indicating activation of the PERK-eIF2-ATF4-CHOP pathway of UPR/ER stress in FECD [17]. In the same study, human CEnCs of FECD tissues demonstrated enlargement/swelling of rough ER compared to control tissues [17]. Jun et al. supported these findings by demonstrating activation of the PERK-ATF4-CHOP ER stress pathways along with swelling of ER in a mouse model of early-onset FECD [18]. However, there are no studies suggesting the direct role of the PERK-eIF2-ATF4-CHOP pathway in inducing morphological changes in ER. Previously, our lab demonstrated increased activation of PERK-ATF4-CHOP as well as IRE-XBP1 in the immortalized human Fuchs cell line (F35T) with 1500 CUG trinucleotide repeats in the transcription factor 4 (TCF4) transcript [19] compared to the normal cell line (21T) [13]. Similarly, Okumura et al. demonstrated that an immortalized FECD cell line had increased expression of CHOP compared to the normal control immortalized endothelial cell line [14]. Okumura et al. also reported increased activation of p-IRE1, consistent with our report of increased expression of its downstream target molecule, X-box binding protein 1 (XBP-1), in immortalized human Fuchs corneal endothelial cell line compared to the control human endothelial cell line [14]. The detailed mechanisms of XBP1 pathways to CEnC apoptosis remain FECD in unclear.

FECD is characterized by corneal guttae, which are outgrowths/excrescences of extracellular matrix (ECM) proteins [20]. Okumura et al. showed the accumulation of ECM proteins, specifically type 1 collagen, fibronectin, and agrin, around aggresomes (thickened Descemet’s membrane (DM) of FECD patients compared to normal control patients [14]. Furthermore, a recent study by Nakagawa et al. found that 32 proteins were expressed in FECD DMs but not in healthy DMs, including hemoglobin α, SRPX2, tenascin-C, and hemoglobin γδεβ, suggesting the distinct composition of guttae compared to healthy DM [20]. Okumura et al. also showed that the use of proteasome inhibitor (MG132)/aggresomes inducer activates the PERK-ATF4-CHOP pathway and mitochondrial-mediated intrinsic apoptotic pathway in the immortalized human corneal cell line [14], suggesting the direct involvement of aggresomes in inducing chronic ER stress and apoptosis. Supporting these findings, Okumura et al. further demonstrated that transforming growth factor-B (TGF-B) is one important protein that regulates excessive ECM protein production, activates the PERK-ATF4-CHOP pathway, and contributes to the activation of the mitochondrial-mediated intrinsic apoptotic pathway in FECD [21]. These studies show the role of excessive ECM deposition in guttae formation and CEnC apoptosis in FECD. However, the detailed mechanisms of how excessive ECM deposition contributes to guttae formation leading to CEnCs apoptosis remain elusive and require further detailed investigation. The summary of ER stress proteins impaired in FECD is described in Table 1.

## 5. Mitochondrial Stress and FECD

In addition to ER stress, mitochondrial stress is another significant factor contributing to the pathogenesis of FECD. CEnCs require many functional mitochondria for ion pump function, which is crucial for maintaining corneal transparency. Mitochondrial stress alters mitochondrial quality control, a complex protective mechanism involving the maintenance of normal mitochondrial energetics and dynamics (balance between fusion and fission), and the clearance of damaged mitochondria by mitophagy. Mitochondrial stress in FECD manifests as changes to mitochondrial bioenergetics and dynamics [23]. Mitochondrial energetics changes in FECD include increased mitochondrial reactive oxygen species (ROS) [24,25,26], decreased ATP production [24], and decreased mitochondrial membrane potential (MMP) [24]. Dysfunctional mitochondrial dynamics in FECD involve decreased mitochondrial fusion with loss of mitofusin 2 (Mfn2), mitochondrial mass [27], and upregulated mitochondrial fission [25]. Although altered mitochondrial bioenergetics and dynamics are known in FECD, their mechanisms for CEnC apoptosis are unclear.

One of the possible mechanisms of the contribution of mitochondrial stress to CEnC apoptosis is the persistent mitochondrial DNA (mtDNA) in FECD. FECD corneal tissue showed telomere shortening and an increased extent of mtDNA damage with lower mitochondrial copy number compared to normal control corneal tissues [28,29]. This aberrant mtDNA damage may be due to an impaired DNA repair process. Ashraf et al. analyzed differential expression profiles of 84 DNA repair genes by real-time PCR and found that FECD specimens showed downregulation of nine and upregulation of eight DNA repair genes compared to specimens from age-matched normal donors [30]. These data suggest mtDNA damage and dysfunctional DNA repair mechanisms in FECD. This mtDNA damage response may further be exacerbated by the oxidative stress seen in FECD. Halilovic et al. demonstrated increased production of reactive oxygen species (ROS) by treating an immortalized corneal endothelial cell line (HCEnC-21T) with menadione, which also caused these cells to form rosettes, a characteristic of FECD [24]. This result was further supported by Liu et al. in vivo, where targeted irradiation of mouse corneas with ultraviolet A (UVA) induced ROS production in the aqueous humor and caused corneal endothelial cell loss, with females presenting a more severe FECD phenotype than did males [25]. However, the question remained whether these ROS are predominantly produced in mitochondria, leading to mitochondrial stress and CEnC apoptosis in FECD. This was shown by Halilovic et al. in the same study, where an increase in mitochondrial-derived ROS resulted in mtDNA damage with subsequent disruption of the inner mitochondrial membrane potential (MMP) depletion of ATP, ultimately resulting in CEnCs apoptosis [24].

Mitochondrial stress regulates mitochondrial quality control systems, namely mitochondrial fission, fusion, and mitophagy, which are further modulated by ROS. Benischke et al. demonstrated that there is excessive mitophagy with increased mitochondrial fragmentation, loss of mitochondrial mass and mitofusin 2 (Mfn2), and the formation of autophagosomes/mitophagosomes in FECD tissues [27]. This study demonstrated that mitophagy drives the loss of Mfn2, thereby establishing the relationship between mitophagy and Mfn2 in FECD [27]. However, the detailed mechanisms of altered/excessive mitophagy and its contribution to CEnCs apoptosis remain elusive. Specifically, it remains unclear whether altered mitophagy affects mitochondrial dynamics, i.e. fusion-fission balance, and contributes to CEnC apoptosis. To further understand the mitophagy mechanisms of action in FECD, Miyai et al. demonstrated that FECD ex vivo specimens exhibited accumulation of PINK1 and phospho-Parkin (Ser65) along with loss of total Parkin, suggesting the role of Parkin-mediated mitophagy. They further demonstrated that oxidative stress regulates Parkin-mediated mitophagy along with mitochondrial fragmentation [26]. However, it remained unclear whether mitophagy affects mitochondrial fusion-fission balance and whether altered mitophagy contributes to CEnC apoptosis in FECD. One probable hypothesis for the relationship between altered mitochondrial dynamics and CEnC apoptosis in FECD is that increased mitochondrial fragmentation and excessive mitophagy activation reduce mitochondrial mass, leading to endothelial cell apoptosis as metabolic processes fall short of cellular energy demands.

Another important question remains whether there is any relationship between guttae and mitochondrial health in FECD. The amount of extracellular membrane material collections, called guttae, forming on Descemet’s membrane was found to correlate inversely with mitochondrial health and survival of endothelial cells [31]. Furthermore, the presence of guttae was also found to positively correlate with the level of mitochondrial calcium and apoptotic cells and negatively correlate with the level of mitochondrial mass and membrane potential [31]. Corneal endothelial cells in guttae-rich regions have reduced mitochondrial activity, and ROS was accordingly found to be lowered in these regions; however, this distribution forces cells in guttae-poor regions to increase their ATP production to compensate, thus leading to mitochondrial exhaustion and increased ROS production, fueling endothelial cell loss seen in FECD. This study suggests that the presence of guttae exacerbates the mitochondrial health of surrounding CEnCs in FECD [31]. As discussed earlier, aberrant mitochondrial dynamics and bioenergetics in corneal endothelial cells play an important role in the pathogenesis of FECD. Given that corneal transplantation is the only known curative option for FECD, it has become imperative to explore alternatives for the treatment of FECD. The mitochondria have become a target for experimental FECD treatments. Recently, Methot et al. co-incubated corneal endothelium explants from FECD patients with exogenous mitochondria, thus enabling exogenous mitochondria internalization into FECD cells [32]. This group found that this procedure reduced oxidative stress, increased MMP, and reversed apoptosis in FECD cells. Thus, mitochondrial transplantation is a promising and innovative potential treatment, but more fundamental basic research is required to answer questions of feasibility and the long-term effects of mitochondrial transplantation in FECD. The summary of mitochondrial stress proteins dysfunction in FECD is described in Table 2.

## 6. ER-Mitochondrial Crosstalk

In the last decade, ER-mitochondrial crosstalk has been shown to play critical roles in neurological [33,34], metabolic [35,36], and age-related diseases [37], although such studies in FECD have been very limited. Shyam et al. first demonstrated that mitochondrial ROS activates ER stress in corneal endothelial cells for Congenital Hereditary Endothelial Dystrophy, a rare recessive blinding disease mainly affecting children [38]. Recently, our lab reported a study where we induced ER stress in healthy human corneal endothelial cells by treating the HCEnC-21T cell line with commonly used ER-stress inducer tunicamycin. We found that all three UPR/ER stress pathways were activated, supporting the claim that ER stress contributes to the pathogenesis of FECD [13]. Building on prior studies that established the existence of ER-mitochondria crosstalk in ocular cells [39,40,41], our lab demonstrated that tunicamycin mediates mitochondrial dysfunction in corneal endothelial cells by altering energetics (with loss of MMP and ATP production), as well as mitochondrial dynamics (with increased mitochondrial fission). Tunicamycin treatment activated all 3 ER/UPR pathways, including the proapoptotic ATF4 pathway, and induced CEnC apoptosis [13]. More recently, we also demonstrated that knockdown of ATF4 reduces MMP loss and mitochondrial fragmentation in 21T cell line under tunicamycin-induced chronic ER stress [15]. However, the detailed mechanisms of ATF4-mediated mitochondrial dysfunction and CEnC apoptosis remain elusive. These exciting data suggest the existence of crosstalk between the ER and mitochondria in FECD, which is a novel finding. Here, we propose several hypotheses for the form and function of this crosstalk in FECD. One major avenue for this crosstalk involves previously described MAMs, which are sites of physical contact between ER and mitochondria that regulate many cellular functions [12], including calcium signaling [42] and lipid synthesis [11,43]. Other potential avenues for this crosstalk involve ER or mitochondria proteins affecting ER or mitochondria stress independent of MAMs.

### 6.1. MAM-Dependent ER-Mitochondrial Crosstalk

MAMs are often dysregulated in cardiovascular [44], neurological [45], metabolic diseases [46], cancer [47], and aging [37]. One common mechanism of MAM-dependent crosstalk is facilitated by ER-mitochondria tethering complex proteins, also called MAM proteins, which bind organellar membranes together and form the structural basis of MAMs [48]. Four tether proteins have been identified and shown to facilitate inter-organellar communication at these sites, and they have been extensively studied in neurodegenerative diseases [48]. The first is VAPB-PTPIP51 (Vesicle-associated membrane protein-associated protein B-protein tyrosine phosphatase-interacting protein-51) tether. VAPB is an ER protein that modulates calcium homeostasis through its interactions with the mitochondrial protein PTPIP51, thus promoting ER-mitochondrial crosstalk [49]. There are no reports of VAPB-PTPIP51 tether in FECD. The second tether protein is Fis1-Bap31 (mitochondrial fission protein Fission 1 homologue-B-cell receptor-associated protein 31) [50]. Fis1 interacts with Bap31 to trigger inter-organellar calcium signaling and promote apoptosis [50]. Recently, we have reported activation of Fis1 in the HCEnC-21 cell line after chronic ER stress [13], suggesting a role in mitochondrial fragmentation. However, there are no reports of Fis1-Bap31 tether in corneal endothelial cells. The third complex is IP3R-Grp75-VDAC (inositol 1,4,5-trisphosphate receptor-glucose-regulated protein 75-voltage-dependent anion channel). Mitochondrial membrane protein VDAC interacts with ER calcium release channel IP3R through Grp75, which is necessary for efficient calcium transfer between ER and mitochondria [51]. There are no reports of IP3R-Grp75-VDAC tether in FECD. The last one is the Mfn2 tether that operates at the MAMs, though its role in crosstalk was the subject of scholarly debate. While some have demonstrated that Mfn2 promotes ER-mitochondrial tethering, others have provided evidence suggesting an antagonistic role of Mfn2 in the tethering process [52,53,54]. However, the study by Naon et al. reconfirmed Mfn2 to be an MAM protein. Further research is required to settle the functions of Mfn2 at MAMs. In FECD, a previous study suggested that activation of mitophagy leads to the decline of Mfn2 and mitochondrial mass in FECD, as described earlier [27]. However, there are no studies suggesting Mfn2 to be a MAM protein in CEnCs. Also, there are currently no descriptions of any MAM tether proteins and their contributions to the pathogenesis of FECD. We have demonstrated these most common mitochondria-ER tethering proteins at MAM known in other cell types (Figure 1). Based on ER and mitochondrial stress studies in FECD, we hypothesize that MAMs will be impaired along with disruptions of MAM tether proteins, which will contribute to ER and mitochondrial stress, leading to CEnCs apoptosis in FECD.

Although most MAM proteins are ER-mitochondria tether proteins, some ER or mitochondrial proteins are non-tether proteins that translocate to MAMs and contribute to MAM-dependent ER-mitochondrial crosstalk. Verfaillie et al. showed that ER-stress protein PERK translocates at the MAM and that knocking out PERK leads to ER-mitochondrial uncoupling and decreased crosstalk [55]. They further demonstrated that PERK translocation facilitates ER-mitochondrial ROS communication and contributes to CEnCs apoptosis [55]. In FECD, there are reports of upregulation of PERK in response to chronic ER stress [13,14], as described earlier, and there are no studies about its translocation at MAM or its functions associated with MAM. Another non-tether protein includes mitochondrial ubiquitin ligase (MITOL), which regulates MAM formation via Mfn2 [56]. Research has demonstrated that MITOL is enriched at the MAM sites [57] and that MITOL knockdown leads to MAM disruption and decreased calcium signaling [56]. Moreover, MITOL regulates ER tethering to mitochondria by activating Mfn2 via K192 ubiquitination [56]. There are no reports of Mfn2 ubiquitination by MITOL in FECD. Another non-tether protein is Parkin, which is an E3 ubiquitin ligase protein that plays a role in ER-mitochondrial crosstalk. Parkin enhances and reduces ER-mitochondrial structural interactions when overexpressed and downregulated, respectively [58,59]. Parkin upregulation also leads to increased calcium transfer between ER and mitochondria [58]. These studies suggest a significant role of Parkin in structural and functional organellar coupling. It is still unclear whether Parkin translocates to MAMs and mediates MAM-dependent ER-mitochondrial crosstalk. A previous study suggests that Parkin facilitates ER-mitochondria interactions through the ubiquitination of the Mfn2 [59]. There are no reports of ATF4 or Parkin protein being localized at MAM in CEnCs. However, Parkin-mediated altered mitophagy [26], along with ATF4 activation [15], is reported in FECD, as mentioned earlier. Protein deglycase DJ-1 is another non-tether protein that localizes to the MAM and interacts with the IP3R3-Grp75-VDAC1 tethering complex; loss of this protein reduces ER-mitochondrial association and disturbs mitochondrial function in Parkinson’s disease, indicating a role in ER-mitochondria crosstalk [60]. In FECD, there is decreased expression of DJ-1, which further attenuates after oxidative stress [61]. Moreover, decreased DJ-1 is associated with nuclear factor erythroid 2-related factor 2 (Nrf2) nuclear translocation and increased susceptibility to CEnC apoptosis [61]. There are no studies suggesting the role of DJ-1 at MAM contributing to the pathogenesis of FECD. Although the role of many MAM-associated proteins in FECD remains to be explored, Mfn2, PERK, DJ-1, and Parkin are all candidates that may play a role in MAM-dependent ER-mitochondria crosstalk.

### 6.2. MAM-Independent ER-Mitochondrial Crosstalk

The ER and mitochondria can also participate in crosstalk independent of MAMs, with ER stress proteins activating mitochondrial stress and vice versa. Several ER stress sensor proteins, including ATF4 [62] and PERK [63], alter mitochondrial dynamics and bioenergetics after chronic ER stress independent of MAMs. Mechanistic studies in the livers of patients with alcoholic hepatitis revealed that upregulated ATF4 represses the transcription activity of nuclear respiratory factor 1 (Nrf1), a major transcriptional regulator of mitochondrial biogenesis, attenuating mitochondrial damage [64]. In the same study, ATF4 ablation in hepatocytes restored previously alcohol-impaired mitochondrial biogenesis and respiratory function [64]. In another study of A549 and H1299 cells, the toxic compound chromium activated ATF4, which resulted in mitophagy, suggesting ER-mitochondria crosstalk [65]. In terms of ATF4’s role in mitochondrial dysfunction in FECD, our lab recently showed that ATF4 knockdown after tunicamycin-induced ER stress rescued altered mitochondrial bioenergetics and dynamics and decreased CEnCs apoptosis, as described earlier [15]. Thus, we suggest that ATF4 of the PERK-ATF4-CHOP pathway is involved in mitochondrial dysfunction, contributing to CEnC apoptosis independent of MAMs. However, we do not rule out the possibility of ATF4 mediating the dysfunction of MAMs in FECD, which requires a detailed investigation. 

Another important molecule of the PERK-ATF4-CHOP pathway is PERK, which interestingly promotes protective stress-induced mitochondrial hyperfusion (SIMH) after ER stress in mouse embryonic fibroblasts [66], suggesting MAM-independent ER-mitochondrial crosstalk. Specifically, PERK signaling remodels mitochondrial phosphatidic acid distribution, leading to the accumulation of phosphatidic acid on the outer mitochondrial membrane and eventual mitochondrial elongation. This phenomenon further prevents mitochondrial fragmentation and promotes mitochondrial metabolism [63]. Moreover, this PERK-mediated ER-mitochondria crosstalk was found to be dependent on eIF2α phosphorylation-dependent translational attenuation [66]. There are no studies suggesting the role of PERK in preventing mitochondrial fragmentation or mediating any aspects of MAM-dependent or independent ER-mitochondria crosstalk in FECD.

ATF6 is another ER stress protein that affects mitochondria function. In response to chronic ER stress, activated ATF6 regulates the expression of Orphan nuclear receptor (ERRγ) [67] and Peroxisome proliferator-activated receptor gamma coactivator 1-alpha (PGC-1α) [68], both implicated in mitochondrial biogenesis. Furthermore, it was recently found that loss of ATF6 in achromatopsia (cone photoreceptor dysfunction) [69] led to mitochondrial respiratory complex dysregulation and altered mitochondrial morphology in retinal organoids [70]. Given that ATF6 plays a role in this ocular disease, future research is required to investigate the role of ATF6 in mitochondria dysfunction seen in FECD. We described ER stress-induced mitochondrial dysfunction independent of MAMs above. However, several mitochondrial proteins, including mitochondrial stress protein, heat shock protein 60 (HSP60) [71], and A-kinase anchoring protein 1 (Akap1) [72], induce ER stress, suggesting a mitochondrial stress-mediated ER response independent of MAMs. In the hepatocytes of C57/B6 mice, overexpression of HSP60 increased ER stress levels, demonstrated by increased CHOP and GRP78 expression [71]. In the same study, ER stress-induced HSP60 expression also impaired mitochondrial function, which suggests that HSP60 is a bidirectional player in ER-mitochondria crosstalk [71]. Another example of mitochondria-ER crosstalk involves Akap1, an important mitochondrial protein needed for mitochondrial homeostasis [72]. In this study, Akap1-/-mice exposed to hyperoxia-induced acute lung injury demonstrated increased ER stress, evident by increased expression of binding immunoglobulin protein (BiP) and eIF2α. This ultimately leads to apoptosis by activating c-Jun-NH (2)- terminal kinase (JNK) pathways [72]. There are several examples of ER-mitochondria crosstalk independent of MAMs, including ER stress proteins affecting mitochondrial function or vice versa. However, we do not rule out the possibility that these above-described ER and mitochondrial stress sensors will mediate ER-mitochondria crosstalk independent of MAMs. Moreover, there are a very limited number of studies [15,38] suggesting MAM-independent ER-mitochondria crosstalk in FECD. We have described the summary of ER-mitochondria crosstalk along with probable proteins that could be associated with MAMs in FECD (Figure 2).

## 7. Conclusions

While previous studies have examined how ER and mitochondrial stress independently contribute to the pathogenesis of FECD, it is less understood how ER-mitochondria crosstalk influences FECD development. Our lab has begun to address this gap in the literature by showing how induction of ER stress leads to mitochondrial dysfunction that results in pathological features seen in FECD. However, the role of ER-mitochondrial interaction in FECD requires further investigation. In particular, we believe that MAMs may play an important role in ER-mitochondrial crosstalk and that disruptions of MAMs and MAM proteins may play a role in FECD development, as seen in other disease states. In the future, it will be beneficial to examine the role played by MAM-dependent as well as independent ER-mitochondrial crosstalk to better understand how each factor contributes to the development of this FECD pathophysiology.

## Figures and Tables

**Figure 1 ijms-26-00894-f001:**
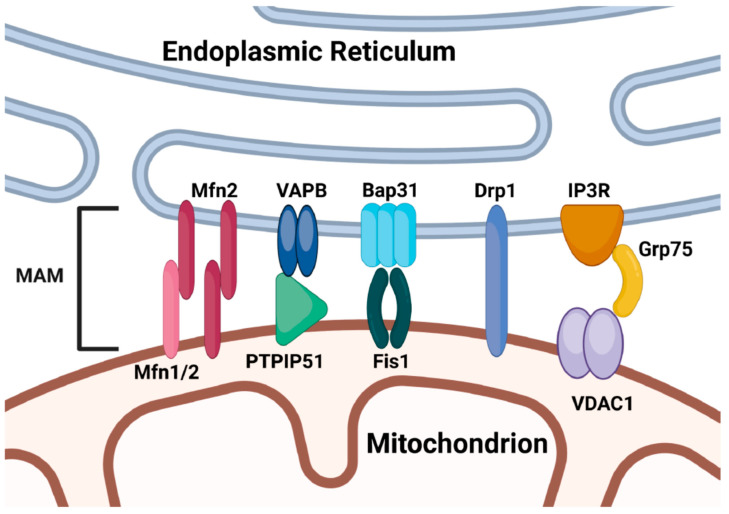
Schematic diagram of ER-mitochondria tethering proteins at the MAM. The diagram demonstrates the structure of major tethering proteins/complexes that mediate the contact site formation between ER and mitochondria. Mitofusin 2 (Mfn2) at the endoplasmic reticulum (ER) interacts with mitochondrial Mitofusin 1 (Mfn1) and Mfn2. ER Vesicle-Associated Membrane Protein (VAPB) interacts with mitochondrial Protein Tyrosine Phosphatase Interacting Protein 51 (PTPIP51). ER B-cell Receptor-Associated Protein (Bap31) interacts with mitochondrial Bap31 Fission-1 Homolog (Fis1). ER Dynamin-Related Protein 1 (Drp1) connects directly to both membranes. ER—Inositol-1,4,5-Triphosphate Receptor (IP3R) and mitochondrial Voltage-Dependent Anion Channel 1 (VDAC1) interact via Glucose-regulated protein 75 (Grp75). Created in https://BioRender.com (accessed on 17 January 2025).

**Figure 2 ijms-26-00894-f002:**
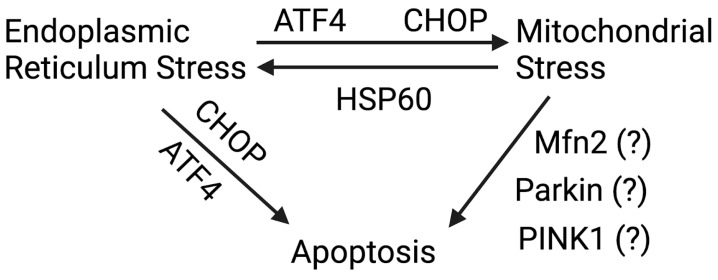
Summary of ER-mitochondria crosstalk in FECD. ER stress pathways (PERK-ATF4-CHOP, IRE-XBP1, and ATF6) are activated in FECD. ER stress response molecules ATF4 and CHOP contribute to mitochondrial stress, leading to loss of MMP, decreased ATP production, and mitochondria fragmentation in FECD. Knockdown of ATF4 or CHOP increases CEnC survival and reduces mitochondrial-mediated intrinsic apoptotic pathway molecules. With respect to mitochondrial stress, proteins such as Mfn2, Parkin, PINK1, etc., are implicated in mitochondrial quality control in FECD; the specific role of these proteins in terms of their contribution to CEnC apoptosis is still unclear and requires in-depth investigation. The question mark (?) indicates that the role of these proteins (Mfn2, Parkin, and PINK1) in apoptosis is unknown. Created in https://BioRender.com (accessed on 5 January 2025).

**Table 1 ijms-26-00894-t001:** Summary of ER stress proteins dysregulated in FECD.

Protein	Model	Role in Apoptosis	Role in FECD	Reference
PERK	Immortalized human CEnCs from FECD	Apoptosis inducer	Upregulated	[14]
IRE1α	Immortalized human CEnCs from FECD	Apoptosis inducer	Upregulated	[14]
CHOP	Human FECD specimens	Apoptosis inducer	Upregulated	[17]
	Immortalized human CEnCs from FECD			[14]
	Normal HCEnC-21T cells treated with ER stressor tunicamycin			[13]
	L450W, Q455K *Col8a2* Knock-In Mouse Models of Fuchs Endothelial Corneal Dystrophy			[22]
eIF2α	Human FECD specimens	Apoptosis inducer	Upregulated	[17]
	HCEnC-21T cells treated with tunicamycin			[13]
GRP78	Human FECD specimens	Apoptosis inducer	Upregulated	[17]
	Immortalized human CEnCs from FECD			[13,14]
	L450W, Q455K *Col8a2* Knock-In Mouse Models of Fuchs Endothelial Corneal Dystrophy			[22]
XBP1	HCEnC-21T cells treated with tunicamycin and Immortalized human CEnCs from FECD	Apoptosis inducer	Upregulated	[13]

**Table 2 ijms-26-00894-t002:** Summary of mitochondria stress proteins dysregulated in FECD.

Protein	Model	Role in Fuchs	Reference
Mfn2	Human FECD specimens, Human FECD cell lines, normal and FECD cell lines treated with mitochondrial depolarization agent, Carbonyl cyanide m-chlorophenyl hydrazone (CCCP)	Downregulated and involved in altered mitochondria quality control/mitophagy in FECD	[27]
Parkin	Human FECD specimens, Normal HCEnC-21T cell line treated with oxidative stress inducer, menadione and Human FECD cell line treated with CCCP	Upregulated and implicated in altered mitophagy in FECD	[26]
Fis-1	Normal HCEnC-21T cell line treated with tunicamycin	Upregulated and contributes to mitochondrial fragmentation in FECD	[13]
Drp1	HCEnC-21T cell line treated with tunicamycin	Upregulated and contributes to mitochondrial fragmentation in FECD	[13]
	Human FECD specimens	Upregulated and contributes to altered mitochondria quality control/dynamics in FECD	[26]
PINK1	Human FECD specimens	Upregulated and involved in altered mitochondria quality control/mitophagy in FECD	[26]

## Data Availability

Not applicable.

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
