# Peer review of "Endoplasmic Reticulum-Mitochondria Crosstalk in Fuchs Endothelial Corneal Dystrophy: Current Status and Future Prospects"

_ijms, 2025, doi:10.3390/ijms26030894_

Round 1
Reviewer 1 Report
Comments and Suggestions for Authors
Kasi and colleagues present a detailed review of the potential molecular mechanisms underlying Fuchs endothelial corneal dystrophy (FECD).
First, they discuss the publications indicating that FECD is dependent on the PERK-ATF4-CHOP pathway activated by endoplasmic reticulum (ER) stress.
Next, the authors review the data indicating the connection between FECD and mitochondrial stress, which is manifested by mitofission and abnormal mitophagy.
Finally, the focus of the review shifts to the connections between mitochondrial and ER stresses determined by both direct contacts and indirect communications between ER and mitochondria.
The review is well-written and is of high interest for both professional ophthalmologists and biomedical researches dealing with the abnormalities of endothelial cell
Suggestions:
1. When speaking about mitochondrial stress, the authors mention “persistent mtDNA in FECD”. What do they mean? mtDNA in cytosol? released mtDNA? Both can induce Inflammatory response, although through different mechanisms. This needs to be clarified.
2. A more detailed description of the extracellular guttae needs to be provided.
3. A graphic scheme showing the proteins involved in ER-mitochondria contacts should be presented.
Author Response
Kasi and colleagues present a detailed review of the potential molecular mechanisms underlying Fuchs endothelial corneal dystrophy (FECD). First, they discuss the publications indicating that FECD is dependent on the PERK-ATF4-CHOP pathway activated by endoplasmic reticulum (ER) stress. Next, the authors review the data indicating the connection between FECD and mitochondrial stress, which is manifested by mito fission and abnormal mitophagy. Finally, the focus of the review shifts to the connections between mitochondrial and ER stresses determined by both direct contacts and indirect communications between ER and mitochondria. The review is well-written and is of high interest for both professional ophthalmologists and biomedical researches dealing with the abnormalities of endothelial cell.
- When speaking about mitochondrial stress, the authors mention “persistent mtDNA in FECD”. What do they mean? mtDNA in cytosol? released mtDNA? Both can induce Inflammatory response, although through different mechanisms. This needs to be clarified.
Persistent mtDNA damage in Fuchs has been measured by long-extension PCR. Persistent mtDNA damage in Fuchs suggests the presence of mitochondrial lesions leading to less amplification of the mtDNA fragment in long-amplicon PCR. This suggests a lower mitochondrial copy number. Fuchs specimens demonstrated a 37% lower mitochondrial copy number than normal age-matched donor corneas.1 Another study demonstrated a significantly increased mtDNA damage with a higher ratio of the common 4977 base pair deletion in the Desmet’s membrane of Fuchs patients compared to the control.2
There is no concept of mtDNA damage in the cytosol or released mtDNA in Fuchs. It is still unclear whether these mtDNA damages can induce inflammation changes in Fuchs. Studies suggest that persistent mtDNA contributes to the pathophysiology of Fuchs.3,4
Now, we have given information about the mtDNA copy number in the revised version.
- A more detailed description of the extracellular guttae needs to be provided.
We have added more information about guttae, describing the results of a recent study using shotgun proteomics to explore its molecular composition.
- A graphic scheme showing the proteins involved in ER-mitochondria contacts should be presented.
Now, we have demonstrated the major Mitochondria-ER tethering proteins at MAM involved in establishing ER-mitochondria contact sites (Figure 1).
References:
- Halilovic, A., Schmedt, T., Benischke, A.S., Hamill, C., Chen, Y., Santos, J.H., and Jurkunas, U.V. (2016). Menadione-Induced DNA Damage Leads to Mitochondrial Dysfunction and Fragmentation During Rosette Formation in Fuchs Endothelial Corneal Dystrophy. Antioxid Redox Signal 24, 1072-1083. 10.1089/ars.2015.6532.
- Czarny, P., Seda, A., Wielgorski, M., Binczyk, E., Markiewicz, B., Kasprzak, E., Jimenez-Garcia, M.P., Grabska-Liberek, I., Pawlowska, E., Blasiak, J., et al. (2014). Mutagenesis of mitochondrial DNA in Fuchs endothelial corneal dystrophy. Mutat Res 760, 42-47. 10.1016/j.mrfmmm.2013.12.001.
- Ashraf, S., Deshpande, N., Vasanth, S., Melangath, G., Wong, R.J., Zhao, Y., Price, M.O., Price, F.W., Jr., and Jurkunas, U.V. (2023). Dysregulation of DNA repair genes in Fuchs endothelial corneal dystrophy. Exp Eye Res 231, 109499. 10.1016/j.exer.2023.109499.
- Czarny, P., Kasprzak, E., Wielgorski, M., Udziela, M., Markiewicz, B., Blasiak, J., Szaflik, J., and Szaflik, J.P. (2013). DNA damage and repair in Fuchs endothelial corneal dystrophy. Mol Biol Rep 40, 2977-2983. 10.1007/s11033-012-2369-2.
- Chen, C.W., Guan, B.J., Alzahrani, M.R., Gao, Z., Gao, L., Bracey, S., Wu, J., Mbow, C.A., Jobava, R., Haataja, L., et al. (2022). Adaptation to chronic ER stress enforces pancreatic beta-cell plasticity. Nat Commun 13, 4621. 10.1038/s41467-022-32425-7.
- Ma, Y., Shimizu, Y., Mann, M.J., Jin, Y., and Hendershot, L.M. (2010). Plasma cell differentiation initiates a limited ER stress response by specifically suppressing the PERK-dependent branch of the unfolded protein response. Cell Stress Chaperones 15, 281-293. 10.1007/s12192-009-0142-9.
- Engler, C., Kelliher, C., Spitze, A.R., Speck, C.L., Eberhart, C.G., and Jun, A.S. (2010). Unfolded protein response in fuchs endothelial corneal dystrophy: a unifying pathogenic pathway? Am J Ophthalmol 149, 194-202 e192. 10.1016/j.ajo.2009.09.009.
- Okumura, N., Kitahara, M., Okuda, H., Hashimoto, K., Ueda, E., Nakahara, M., Kinoshita, S., Young, R.D., Quantock, A.J., Tourtas, T., et al. (2017). Sustained Activation of the Unfolded Protein Response Induces Cell Death in Fuchs' Endothelial Corneal Dystrophy. Invest Ophthalmol Vis Sci 58, 3697-3707. 10.1167/iovs.16-21023.
- Qureshi, S., Lee, S., Steidl, W., Ritzer, L., Parise, M., Chaubal, A., and Kumar, V. (2023). Endoplasmic Reticulum Stress Disrupts Mitochondrial Bioenergetics, Dynamics and Causes Corneal Endothelial Cell Apoptosis. Invest Ophthalmol Vis Sci 64, 18. 10.1167/iovs.64.14.18.
- Qureshi, S., Lee, S., Ritzer, L., Kim, S.Y., Steidl, W., Krest, G.J., Kasi, A., and Kumar, V. (2024). ATF4 regulates mitochondrial dysfunction, mitophagy, and autophagy, contributing to corneal endothelial apoptosis under chronic ER stress in Fuchs dystrophy. bioRxiv, 2024.2011.2014.623646. 10.1101/2024.11.14.623646.
- Hu, J., Rong, Z., Gong, X., Zhou, Z., Sharma, V.K., Xing, C., Watts, J.K., Corey, D.R., and Mootha, V.V. (2018). Oligonucleotides targeting TCF4 triplet repeat expansion inhibit RNA foci and mis-splicing in Fuchs' dystrophy. Hum Mol Genet 27, 1015-1026. 10.1093/hmg/ddy018.

Reviewer 2 Report
Comments and Suggestions for Authors
The document is organized and clear and presents a direct perspective, but in order to homogenize it is recommended to eliminate the names of tables 1 and 2, leaving only the number of their references.
The work is a perspective that puts into context the role of FECD in Crosstalk Endoplasmic Corneal Dyscracy, as well as some future prospects, making clear the relationship of several pathways involved in PERK and mitochondrial stress, and also proposes an associated mechanism for this pathology, presenting clear and concrete conclusions generating an integration of everything analyzed in the perspective. There are minimal errors that need correction in the use of italics in terms of in vivo and ex vivo, as well as the homogenization of the document. In tables 1 and 2, use only the reference number, without including the name of the authors. The work puts the subject into context for the scientific community. giving way to other studies to emphasize the work,
its publication could be approved in this state, however, minor corrections are considered necessary to homogenize the tables and the use of italics in terms such as in vivo and ex vivo (line 178)
Author Response
The work is a perspective that puts into context the role of FECD in Crosstalk Endoplasmic Corneal Dystrophy, as well as some prospects, making clear the relationship of several pathways involved in PERK and mitochondrial stress, and also proposes an associated mechanism for this pathology, presenting clear and concrete conclusions generating an integration of everything analyzed in the perspective. There are minimal errors that need correction in the use of italics in terms of in vivo and ex vivo, as well as the homogenization of the document. In Tables 1 and 2, use only the reference number without including the name of the authors. The work puts the subject into context for the scientific community giving way to other studies to emphasize the work.
Its publication could be approved in this state. However, minor corrections are considered necessary to homogenize the tables and the use of italics in terms such as in vivo and ex vivo (line 178)
In Tables 1 and 2, the authors’ names have been removed. Only reference numbers are provided. We have ensured that words like in vivo and ex vivo are italicized.

Reviewer 3 Report
Comments and Suggestions for Authors
Kasi et al. review the state of the art in Fuchs endothelial corneal dystrophy (FECD) with a focus on ER-mitochondrial crosstalk and their own work. FECD is a quite prevalent, debilitating disease, therefore the topic is highly interesting and relevant. The review is concise and well written but before recommendation for publication I´d like to have some comments and suggestions addressed.
1. line 66. Prolonged/chronic ER stress….Is that so? Imho only unresolved ER stress leads to apoptosis, I would think there are conditions of prolonged ER stress (plasma cells, pancreatic cells, etc.) that do not result in apoptosis.
2. Citation #13 is wrong, I didn´t find anything in there related to accumulation of misfolded proteins. What is meant, accumulation in the ER, triggering UPR? This should result in less secretion. Extracellular accumulation? Does this mean secretion is actually increased in FECD? Please check citation and better explain. Please also check the other citations for accuracy.
3. line 86. Sorry, but I think citing your work on HCEnCs here is irrelevant. #17 shows that HCEnCs are capable of reacting to tunicamycin, but every cell does that. This does not provide any clues what is going on in FECD. Are HCEnCs per se more susceptible for ER stress compared to other cells, fibroblasts, others? Is there more reason for ER stress in FECD (misfolded proteins? Why?) or did they get more susceptible (why?). The increased PERK-CHOP pathway in F35T cells is relevant, but so is the increase in XBP, why do you focus on PERK? ATF6 unfortunately was not analyzed in F35T cells.
4. line 89.”having 1500 repeats”. Of what?
5. line 96-113. Again I don´t find this disease-relevant, these are all normal reactions to ER-stress inducing agents with no connection to FECD.
6. line 114-. A general consideration: Could it be that the ER stress is secondary, because for some reasons secretion in FECD is upregulated, explaining the accumulation of ECM proteins? Such an increased secretion would automatically induce UPR. For some reason the UPR can not cope with the increased secretion, leading to PERK activation and apoptosis.
7. line 218- You mentioned that before, no need to repeat here.
8. line 222. Do you really need to underline the first? And please be precise, you were not the first to show that connection, that was show 20 years ago. Maybe you are the first to show it in CEnC, but then please state that and acknowledge your colleagues that published that using other cells.
9. Fig. 1. From the text it seems to me there should also be an arrow labelled HSP60 back from mito stress to ER stress, no?
Author Response
Kasi et al. review the state of the art in Fuchs endothelial corneal dystrophy (FECD) with a focus on ER-mitochondrial crosstalk and their own work. FECD is a quite prevalent, debilitating disease. Therefore, the topic is highly interesting and relevant. The review is concise and well-written, but before the recommendation for publication, I´d like to have some comments and suggestions addressed.
- line 66. Prolonged/chronic ER stress….Is that so? Imho only unresolved ER stress leads to apoptosis, I would think there are conditions of prolonged ER stress (plasma cells, pancreatic cells, etc.) that do not result in apoptosis.
Prolonged/severe ER stress often leads to apoptosis, and this concept is true in most cases. However, the specific apoptotic pathways and sensitivity to apoptosis might differ between cell types based on their unique protein synthesis demands. For example, Chen et al. demonstrated pancreatic B cell undergo plasticity as an adaptive response to chronic ER stress.5 Specifically, b cells experience a homeostatic change associated with the reprogramming of their transcriptome and translatome, thereby compromising their identity. However, upon relief from chronic ER stress, b cells regain their identity, suggesting their plasticity. Similarly, plasma cells regulate different branches of UPR differently to accomplish distinct outcomes using the same UPR machinery.6
Previous studies7,8, as well as our study9, have seen unresolved ER stress in Fuchs, which is implicated in corneal endothelial apoptosis. To address this point, we have changed the word “prolonged” to “unresolved.” No studies suggest the role of separate role of acute and chronic ER stress in Fuchs
- Citation #13 is wrong, I didn´t find anything in there related to accumulation of misfolded proteins. What is meant, accumulation in the ER, triggering UPR? This should result in less secretion. Extracellular accumulation? Does this mean secretion is actually increased in FECD? Please check citation and better explain. Please also check the other citations for accuracy.
We have removed citation 13 and placed it in the section “mitochondria stress and Fuchs.” This paper describes how guttae affect mitochondrial health, contributing to the loss of corneal endothelial cells. We have also checked all of our other citations for accuracy.
3.line 86. Sorry, but I think citing your work on HCEnCs here is irrelevant. #17 shows that HCEnCs are capable of reacting to tunicamycin, but every cell does that. This does not provide any clues what is going on in FECD. Are HCEnCs per se more susceptible for ER stress compared to other cells, fibroblasts, others? Is there more reason for ER stress in FECD (misfolded proteins? Why?) or did they get more susceptible (why?). The increased PERK CHOP pathway in F35T cells is relevant, but so is the increase in XBP, why do you focus on PERK? ATF6 unfortunately was not analyzed in F35T cells.
We agree that the corneal endothelial cell line's response to tunicamycin is similar to that of any other cell line, and this has nothing to do with Fuchs. Now, we have made a separate paragraph, “ER stress and Corneal Endothelial Cells,” before discussing ER stress in Fuchs.
We have now cited our new paper (currently in BioXriv)10 in the section of ER stress in Fuchs, in which we show a differential response of ER stress pathways in normal (21T) and Fuchs (F35T) cell lines after chronic ER stress. We focused on PERK-ATF4 -CHOP pathways in this paper since this pathway is well-studied in apoptosis in other systems. We will explore other ER stress pathways in future.
- line 89.” having 1500 repeats”. Of what?
F35T corneal endothelial cell line is derived from Fuchs patient expressing transcription factor 4 (TCF4) with approximately 1500 CUG repeats.11 We have added this information.
- Lines 96-113. Again, I don´t find this disease-relevant. These are all normal reactions to ER-stress-inducing agents with no connection to FECD.
We have created a separate section, “ER stress and Corneal endothelial cells,” where we describe the ER stress response in corneal endothelial cell lines.
- line 114-. A general consideration: Could it be that the ER stress is secondary, because for some reasons secretion in FECD is upregulated, explaining the accumulation of ECM proteins? Such an increased secretion would automatically induce UPR. For some reason the UPR can not cope with the increased secretion, leading to PERK activation and apoptosis.
Yes, the hypothesis mentioned here regarding ER stress being secondary is possible, though this hypothesis has not been tested in Fuchs.
- line 218- You mentioned that before; no need to repeat here.
This is the first and only mention of congenital hereditary endothelial dystrophy in the paper. We need to keep this.
- line 222. Do you really need to underline the first? And please be precise, you were not the first to show that connection, that was show 20 years ago. Maybe you are the first to show it in CEnC, but then please state that and acknowledge your colleagues that published that using other cells.
We agree with this comment. We were among the first to show ER-mitochondria crosstalk in corneal endothelial cells. We have removed the underlining and referenced our colleagues’ prior work showing ER stress-induced mitochondrial dysfunction in other cell types throughout the paper.
- Fig. 1. From the text, it seems that there should also be an arrow labeled HSP60 back from mito stress to ER stress, no?
We have added Hsp60, put arrows in Figure 1.
References:
- Halilovic, A., Schmedt, T., Benischke, A.S., Hamill, C., Chen, Y., Santos, J.H., and Jurkunas, U.V. (2016). Menadione-Induced DNA Damage Leads to Mitochondrial Dysfunction and Fragmentation During Rosette Formation in Fuchs Endothelial Corneal Dystrophy. Antioxid Redox Signal 24, 1072-1083. 10.1089/ars.2015.6532.
- Czarny, P., Seda, A., Wielgorski, M., Binczyk, E., Markiewicz, B., Kasprzak, E., Jimenez-Garcia, M.P., Grabska-Liberek, I., Pawlowska, E., Blasiak, J., et al. (2014). Mutagenesis of mitochondrial DNA in Fuchs endothelial corneal dystrophy. Mutat Res 760, 42-47. 10.1016/j.mrfmmm.2013.12.001.
- Ashraf, S., Deshpande, N., Vasanth, S., Melangath, G., Wong, R.J., Zhao, Y., Price, M.O., Price, F.W., Jr., and Jurkunas, U.V. (2023). Dysregulation of DNA repair genes in Fuchs endothelial corneal dystrophy. Exp Eye Res 231, 109499. 10.1016/j.exer.2023.109499.
- Czarny, P., Kasprzak, E., Wielgorski, M., Udziela, M., Markiewicz, B., Blasiak, J., Szaflik, J., and Szaflik, J.P. (2013). DNA damage and repair in Fuchs endothelial corneal dystrophy. Mol Biol Rep 40, 2977-2983. 10.1007/s11033-012-2369-2.
- Chen, C.W., Guan, B.J., Alzahrani, M.R., Gao, Z., Gao, L., Bracey, S., Wu, J., Mbow, C.A., Jobava, R., Haataja, L., et al. (2022). Adaptation to chronic ER stress enforces pancreatic beta-cell plasticity. Nat Commun 13, 4621. 10.1038/s41467-022-32425-7.
- Ma, Y., Shimizu, Y., Mann, M.J., Jin, Y., and Hendershot, L.M. (2010). Plasma cell differentiation initiates a limited ER stress response by specifically suppressing the PERK-dependent branch of the unfolded protein response. Cell Stress Chaperones 15, 281-293. 10.1007/s12192-009-0142-9.
- Engler, C., Kelliher, C., Spitze, A.R., Speck, C.L., Eberhart, C.G., and Jun, A.S. (2010). Unfolded protein response in fuchs endothelial corneal dystrophy: a unifying pathogenic pathway? Am J Ophthalmol 149, 194-202 e192. 10.1016/j.ajo.2009.09.009.
- Okumura, N., Kitahara, M., Okuda, H., Hashimoto, K., Ueda, E., Nakahara, M., Kinoshita, S., Young, R.D., Quantock, A.J., Tourtas, T., et al. (2017). Sustained Activation of the Unfolded Protein Response Induces Cell Death in Fuchs' Endothelial Corneal Dystrophy. Invest Ophthalmol Vis Sci 58, 3697-3707. 10.1167/iovs.16-21023.
- Qureshi, S., Lee, S., Steidl, W., Ritzer, L., Parise, M., Chaubal, A., and Kumar, V. (2023). Endoplasmic Reticulum Stress Disrupts Mitochondrial Bioenergetics, Dynamics and Causes Corneal Endothelial Cell Apoptosis. Invest Ophthalmol Vis Sci 64, 18. 10.1167/iovs.64.14.18.
- Qureshi, S., Lee, S., Ritzer, L., Kim, S.Y., Steidl, W., Krest, G.J., Kasi, A., and Kumar, V. (2024). ATF4 regulates mitochondrial dysfunction, mitophagy, and autophagy, contributing to corneal endothelial apoptosis under chronic ER stress in Fuchs dystrophy. bioRxiv, 2024.2011.2014.623646. 10.1101/2024.11.14.623646.
- Hu, J., Rong, Z., Gong, X., Zhou, Z., Sharma, V.K., Xing, C., Watts, J.K., Corey, D.R., and Mootha, V.V. (2018). Oligonucleotides targeting TCF4 triplet repeat expansion inhibit RNA foci and mis-splicing in Fuchs' dystrophy. Hum Mol Genet 27, 1015-1026. 10.1093/hmg/ddy018.

Round 2
Reviewer 1 Report
Comments and Suggestions for Authors
The authors properly answered the critiques